# Diterpene Resin Acids and Olefins in Calabrian Pine (*Pinus nigra* subsp. *laricio* (Poiret) Maire) Oleoresin: GC-MS Profiling of Major Diterpenoids in Different Plant Organs, Molecular Identification and Expression Analysis of Diterpene Synthase Genes

**DOI:** 10.3390/plants10112391

**Published:** 2021-11-05

**Authors:** Enrica Alicandri, Stefano Covino, Bartolomeo Sebastiani, Anna Rita Paolacci, Maurizio Badiani, Francesco Manti, Carmelo Peter Bonsignore, Agostino Sorgonà, Mario Ciaffi

**Affiliations:** 1Dipartimento di Agraria, Università Mediterranea di Reggio Calabria, Loc. Feo di Vito, 89129 Reggio Calabria, Italy; e.alicandri@gmail.com (E.A.); mbadiani@unirc.it (M.B.); asorgona@unirc.it (A.S.); 2Dipartimento per la Innovazione nei Sistemi Biologici, Agroalimentari e Forestali, Università della Tuscia, Via S. Camillo De Lellis, s.n.c., 01100 Viterbo, Italy; stefano.covino80@gmail.com (S.C.); arpaolacci@unitus.it (A.R.P.); 3Dipartimento di Chimica, Biologia e Biotecnologie, Università di Perugia, Via Elce di Sotto 8, 06123 Perugia, Italy; bartolomeo.sebastiani@unipg.it; 4Dipartimento di Patrimonio, Architettura e Urbanistica, Università Mediterranea di Reggio Calabria, Salita Melissari, 89124 Reggio Calabria, Italy; francesco.manti@unirc.it (F.M.); cbonsignore@unirc.it (C.P.B.)

**Keywords:** diterpenoids, diterpene synthase, *Pinus nigra* subsp. *laricio* (Poiret) Maire, Calabrian pine, pine oleoresin, genomic organization of gymnosperm diterpene synthases, diterpene resin acid

## Abstract

A quali-quantitative analysis of diterpenoid composition in tissues obtained from different organs of *Pinus nigra* subsp. *laricio* (Poiret) Maire (Calabrian pine) was carried out. Diterpene resin acids were the most abundant diterpenoids across all the examined tissues. The same nine diterpene resin acids were always found, with the abietane type prevailing on the pimarane type, although their quantitative distribution was found to be remarkably tissue-specific. The scrutiny of the available literature revealed species specificity as well. A phylogeny-based approach allowed us to isolate four cDNAs coding for diterpene synthases in Calabrian pine, each of which belonging to one of the four groups into which the d3 clade of the plants’ terpene synthases family can be divided. The deduced amino acid sequences allowed predicting that both monofunctional and bifunctional diterpene synthases are involved in the biosynthesis of diterpene resin acids in Calabrian pine. Transcript profiling revealed differential expression across the different tissues and was found to be consistent with the corresponding diterpenoid profiles. The isolation of the complete genomic sequences and the determination of their exon/intron structures allowed us to place the diterpene synthase genes from Calabrian pine on the background of current ideas on the functional evolution of diterpene synthases in Gymnosperms.

**Key** **to** **Plant** **Species:***Abies balsamea* (L.) Mill., 1768; balsam fir; *Abies grandis* (Douglas ex D. Don) Lindl., grand fir; *Physcomitrella patens* (Hedw.) Bruch & Schimp., spreading earthmoss; *Picea abies* (L.) H. Karst; Norway spruce; *Picea glauca* (Moench) Voss; white spruce; *Picea sitchensis* (Bongard) Carrière; 1855; Sitka spruce; *Pinus banksiana* Lamb., jack pine; *Pinus contorta* Douglas; lodgepole pine; *Pinus nigra* J.F. Arnold; Austrian pine or black pine; *Pinus nigra* subsp. *laricio* (Poiret) Maire; Calabrian pine; *Pinus pinaster* Aiton; maritime pine; *Pinus radiata* D. Don; Monterey pine; *Pinus taeda* L., loblolly pine; *Pseudolarix amabilis* (N. Nelson) Rehder; golden larch.

## 1. Introduction

Gymnosperms developed a variety of physical and chemical defences against pathogens and herbivores, among which one of the most significant is the production of terpenoid metabolites [1,2,3,4]. The complex terpenoid defence mechanisms have persisted throughout the long evolutionary history of gymnosperms and their decreasing geographical distribution during the Cenozoic era [5,6], but diversified into often species-specific metabolite blends. For instance, structurally related labdane-type diterpenoids, such as ferruginol and derivative compounds, act as defence metabolites in many Cupressaceae species [3,7,8]. On the other hand, diterpene resin acids (DRAs), together with mono- and sesqui-terpenes, are the main components of the oleoresin defence system in the Pinaceae species (e.g., conifers), and have been shown to provide an effective barrier against stem-boring weevils and associated pathogenic fungi [9,10,11,12].

Diterpenoids from gymnosperms are also important for their technological uses, being employed in the production of solvents, flavours, fragrances, pharmaceuticals and a large selection of bioproducts [1,13], such as, among the many other examples, the anticancer drugs pseudolaric acid B, obtained from the roots of the golden larch (*Pseudolarix amabilis*) [14], and taxol, extracted from yew (*Taxus* spp.) [15], as well as *cis*-abienol, produced by balsam fir (*Abies balsamea*), which is a molecule of interest for the fragrance industry [16].

The diterpenoids of conifer oleoresin are largely members of three structural groups: the abietanes, the pimaranes, and the dehydroabietanes, all of which are characterized by tricyclic parent skeletons [2,17]. These diterpenoids are structurally similar to the tetracyclic *ent*-kaurane diterpenes, which include the ubiquitous gibberellin (GA) phytohormones. Both the oleoresin diterpenoids of specialized metabolism and the GAs of general metabolism derive from the common non-cyclic diterpenoid precursor geranylgeranyl diphosphate (GGPP).

In conifers, among the other gymnosperms, the structural diversity of diterpenoids results from the combined actions of diterpene synthases (DTPSs) and cytochrome P450 monooxygenases (CP450s) [2]. The former enzymes catalyse the cyclization and rearrangement of the precursor molecule GGPP into a range of diterpene olefins, often referred to as the neutral components of the oleoresins. Olefins are then functionalized at specific positions by the action of CP450s, through a sequential three-step oxidation first to the corresponding alcohols, then to aldehydes, and finally to DRAs [2], such as abietic, dehydroabietic, isopimaric, levopimaric, neoabietic, palustric, pimaric, and sandaracopimaric acids, which are the major constituents of conifer oleoresins [2,17,18]. The chemical structures of the most-represented diterpenoids in *Pinus* spp. are reported in Appendix A.

Diterpene synthases in gymnosperms share a conserved α-helical fold with a common three-domain γβα architecture, and characteristic functional motifs (DxDD, DDxxD, NSE/DTE), which determine the catalytic activity of the enzymes [18,19]. Indeed, depending on domain structure and presence/absence of signature active-site motifs, three major classes of DTPSs can be identified, namely monofunctional class I and class II DTPSs (mono-I-DTPS and mono-II-DTPS in the following, respectively) and bifunctional class I/II DTPSs (bi-I/II-DTPSs in the following) [20]. Mono-II-DTPSs contain a conserved DxDD motif located at the interface of the γ and β domains, which is critical for facilitating the protonation-initiated cyclization of GGPP into bicyclic prenyl diphosphate intermediates [21], among which copalyl diphosphate (CPP) and labda-13-en-8-ol diphosphate (LPP) are the most common [3,22,23]. Mono-I-DTPSs then convert the above bicyclic intermediates into the tricyclic final structures, namely diterpene olefins, by ionization of the diphosphate group and rearrangement of the carbocation, which is facilitated by a Mg^2+^ cluster coordinated between the DDxxD and the NSE/DTE motifs in the C-terminal α-domain.

Bi-I/II-DTPSs, regarded as the major enzymes involved in the specialized diterpenoid metabolism in conifers, contain all the three functional active sites, namely DxDD (between γ and β domains), DDxxD and NSE/DTE (in the α-domain), and therefore are able to carry out in a single step the conversion of the linear precursor GGPP into the final tricyclic olefinic structures, which serve in turn as the precursors for the most abundant DRAs in each species [24]. In contrast, the synthesis of GA precursor *ent*-kaurene in gymnosperms involves two consecutively acting mono-I- and mono-II-DTPSs, namely *ent*-CPP synthase (*ent*-CPS) and *ent*-kaurene synthase (*ent*-KS), respectively, as has also been shown for both general and specialized diterpenoid metabolism in angiosperms [18,20,25]. Interestingly, class-I DTPSs involved in specialized diterpenoid metabolism were identified in *Pinus contorta* and *Pinus banksiana*, which can convert (+)-CPP produced by bifunctional DTPSs to form pimarane-type diterpenes [22], while no (+)-CPP producing class-II DTPSs have been identified in other conifers.

Most of the existing knowledge concerning the genetics and metabolism of specialized diterpenes in gymnosperms was obtained from model Pinaceae species, such as *Picea glauca*, *Abies grandis*, *Pinus taeda,* and *P. contorta* [1,2,22], for which large transcriptomic and genomic resources are available, as well as, in recent times, from species occupying key position in the gymnosperm phylogeny, such as those belonging to the Cupressaceae and the Taxaceae families [3,23]. In previous works of ours [20,26], we began to gain insight into the ecological and functional roles of the terpenes produced by the non-model conifer *Pinus nigra* subsp. *laricio* (Poiret) (Calabrian pine), one of the six subspecies of *P. nigra* (black pine) and an insofar completely neglected species under such respect. In terms of natural distribution, black pine is one of the most widely distributed conifers over the whole Mediterranean basin, and its *laricio* subspecies is considered endemic of southern Italy, especially of Calabria, where it is a basic component of the forest landscape, playing key roles not only in soil conservation and watershed protection, but also in the local forest economy [27].

In the present study, we carried out for the first time, to the best of our knowledge, a quali-quantitative analysis of diterpenoids composition in different tissues of Calabrian pine via conventional gas chromatography-mass spectrometry (GC-MS). In this same subspecies, in addition, we report here about the isolation of full length (FL) cDNAs and the corresponding genomic sequences encoding for DTPSs involved in the specialized diterpenoid metabolism, obtained by using a strategy based on the phylogeny of available DTPSs from different *Pinus* species. The isolation of DTPS genes made a tissue-specific gene expression analysis possible, to be confronted with the corresponding GC-MS diterpene profiles.

## 2. Results and Discussion

### 2.1. In the Pinaceae, the Diterpene Metabolites Profiles Are Tissue-Specific and Species-Specific

The diversity of oleoresin diterpenoids and the extent of diterpene oxidation were quali-quantitatively evaluated in five different Calabrian pine tissues, namely young (YN) and mature (MN) needles, bark and xylem combined from leader (LS) and interwhorl (IS) stems, and roots (R).

GC-MS analysis showed that diterpene resin acids (DRAs) are the most abundant diterpenoids across all the examined tissue types, together with remarkably lower amounts of the corresponding aldehydes and olefins (Appendix A). Similar quantitative relationships among acidic and neutral diterpenoids were previously observed in various tissue types of other *Pinus* species, such as *P. banksiana* and *P. contorta* [22], as well as *P. pinaster* and *P. radiata* [28]. Likewise, in Sitka spruce (*Picea sitchensis*), the DRA fraction in stem tissues accounted for more than 92% of the total diterpenoids [17]. Because of their very low concentrations in all the tissues of Calabrian pine examined, olefins and aldehydes are described here only qualitatively, whereas the corresponding DRAs are quantitatively compared among each other in the different tissues (see below).

All the Calabrian pine tissues examined here showed the presence of the same nine DRAs, seven of which were non-dehydrogenated species—namely pimaric acid, sandaracopimaric acid, isopimaric acid, palustric acid, levopimaric acid, abietic acid, and neoabietic acid—and two being dehydrogenated ones, namely dehydroabietic acid and a non-identified putative dehydroisomer. This is exemplified in Appendix A, showing the DRA elution profiles obtained from the LS tissue and in Appendix A, illustrating their mass spectra.

Quantitatively speaking, Figure 1A shows that the highest contents of total DRAs were found in the LS and IS tissues, with decreasing concentrations being observed in the R, MN and YN ones. Figure 1B–F also shows the quantitative distribution of the nine DRAs in the different tissue examined: in both MN and YN, dehydroabietic, isopimaric and abietic acids were found to be the main components, while the other DRAs were detected at lower concentrations (1–6% of the total). This confirms the results obtained by López-Goldar et al. [28] on the same tissues of *P. radiata* and *P. pinaster*, but not those reported by Hall et al. [22], who instead observed a prevalence of levopimaric and neoabietic acids in both young and mature needles from *P. contorta* and *P. banksiana*. In the LS tissue, abietic acid was the dominant DRA component (about the 33% of the total), followed by dehydroabietic and palustric acids. On the other hand, the IS tissue showed a prevalence of dehydroabietic and palustric acids, each contributing about 30% of the total DRAs, followed by abietic acid. In both the stem tissues, namely LS and IS, comparatively lower abundances were observed for levopimaric, isopimaric, pimaric, sandaracopimaric, and neoabietic acids, as well as for the non-identified dehydroisomer. These results significantly differ from those reported by Hall et al. [22], who instead observed that levopimaric acid is the most abundant DRA in the LS and IS tissues from *P. contorta* and *P. banksiana*. Finally, dehydroabietic, palustric and abietic acids, although with significant differences in their amounts, were found to be the predominant DRAs of the R tissue, in which, compared to the aforementioned aerial tissues, intermediate abundances of isopimaric- and levopimaric acids, as well as lower amounts of pimaric-, sandaracopimaric-, neoabietic acids, and of the non-identified dehydroisomer, were measured. Again differently to our results, Hall et al. [22] reported comparatively higher concentrations of palustric and levopimaric acids in the roots of both *P. contorta* and *P. banksiana*. Taken together, the reported results could suggest that the DRA fingerprint in Pinus spp. is not only tissue-specific, but also species-specific.

In conifer oleoresins, both due to their nature of precursors, and because of their higher volatility and tendency to undergo UV-induced photooxidation, olefins are normally found in lower concentrations with respect to their oxygen-containing counterparts, i.e., DRAs. In agreement with such a view, we detected in all the Calabrian pine tissues only trace amounts of the neutral components of oleoresin, of which there were five olefins, namely sandaracopimaradiene, levopimaradiene, palustradiene, abietadiene, and neoabietadiene, and five aldehydic derivatives, namely sandaracopimaradienal, palustradienal, isopimaradienal, abietadienal, and neoabietadienal (Appendix A). Qualitatively speaking, the olefins and the corresponding aldehydes found in Calabrian pine tissues were the same as those found by Hall et al. [22] in the homologous tissues of *P. contorta* and *P. banksiana*, although at different relative concentrations.

### 2.2. A Phylogeny-Based Approach for Isolating Partial and Full-Length cDNAs Coding for Diterpene Synthases in Calabrian Pine

To gain insight into the structural diversity of diterpenoids in Calabrian pine, we isolated cDNA sequences encoding DTPSs potentially involved in the synthesis of the specialized diterpenes acting as DRA precursors in such species. The strategy adopted was based on the PCR amplification of cDNA sequences by using specific primers designed on conserved regions of pine DTPSs belonging to distinct phylogenetic groups, an approach we successfully used previously for the isolation of genes encoding monoterpene synthases in the same non-model conifer species [20].

In a previous work of ours [20], we carried out an extensive in silico search to identify all the putative full-length TPSs for primary and specialized metabolisms in different *Pinus* species, and to analyze their phylogenetic relationships. As far as DTPSs are concerned, such a database search allowed us to identify 13 FL sequences involved in the secondary diterpenoid metabolism in the *Pinus* species (Appendix A). Phylogenetic analysis clustered all the 13 pine DTPSs sequences into the TPS-d3 clade, which includes four well-supported major groups, denoted as 1–4. Each of these groups contains DTPS proteins from different pine species thought to be functionally related among each other [20].

Based on our previous phylogenetic analysis, in the present work the deduced amino acid and the nucleotide sequences of pine DTPSs belonging to each of the aforementioned groups 1–4 (Appendix A) were aligned, in order to identify highly conserved regions among members of each of the four groups. Such conserved regions within each group were then used to design specific primers for the isolation by RT-PCR of partial transcripts of orthologous genes in Calabrian pine. Appendix A schematically outlines the FL cDNAs for representative members of the four phylogenetic DTPS groups, and the positions of the specific primers used, of which a complete list is reported in Appendix A. By using such a strategy, we were able to isolate and sequence partial DTPS transcripts of putative orthologous genes in Calabrian pine belonging to each of the groups 1–4, which confirms the validity of the phylogenetic approach used. These partial DTPS transcripts were then used as templates for isolating the corresponding FL cDNA sequences by means of 5′ and 3′ RACE (Rapid Amplification of cDNA Ends) extensions; the primer sequences of 5′ and 3′ RACE are reported in Appendix A and their positions indicated in Appendix A.

In the cases of the partial DTPS transcripts belonging to groups 1 and 2, two slightly different sequences were recognized among the three clones analysed for each cDNA fragment due to nucleotide substitutions, most of them synonymous, on a background otherwise showing high levels of sequence identity among each other (over 97%). These slightly different DTPS transcripts might derive from alleles of the same gene and/or from duplicated copies of the same gene, and this would imply that we might have as many more DTPS closely related genes belonging to each phylogenetic group in Calabrian pine, as observed in other *Pinus* species [22]. This possibility will be tested in future studies. However, among the three sequenced clones for the corresponding 3′ and 5′ RACE products, we identified the same sequences that were identical to the 3′ and 5′ ends of two of the three sequenced cDNA products, indicating that they are part of the same FL transcript. Therefore, the assembled four unique FL cDNAs isolated from Calabrian pine, denoted as *Pnl DTPS1*, *Pnl DTPS2*, *Pnl DTPS3*, and *Pnl DTPS4*, each of them belonging to one of the four groups of the TPS-d3 clade, contained open reading frames (ORFs) of 2574, 2559, 2631 and 2607 bp, respectively, and were predicted to encode proteins of 857, 852, 876 and 868 aa, respectively (Figure 2).

The FL cDNA sequences of the *Pnl DTPS*1–4 genes have been deposited in the GeneBank database under the accession numbers OK245418 to OK245421.

### 2.3. Sequence-Based Analysis Predicts That Both Monofunctional and Bifunctional Diterpene Synthases Are Involved in the Biosynthesis of Diterpene Resin Acids in Calabrian Pine

The deduced amino acid sequences of the four full-length cDNAs isolated from Calabrian pine (see above) were found to contain highly conserved and characteristic regions of plant DTPSs (Figure 2). First, a putative transit peptide, ranging from 33 (Pnl DTPS1) to 68 aa (Pnl DTPS4) in length, probably needed for the import of the mature DTPS proteins into plastids. Secondly, DTPS active-site signature motifs (Figure 2): Pnl DTPS1 was found to contain both class-II and class-I motifs, suggesting its nature of proper bi-I/II DTPS, like the already-known bifunctional DTPSs involved in DRAs biosynthesis in conifers, namely the isopimaradiene synthase-type (ISO) and levopimaradiene/abietadiene synthase-type (LAS) enzymes from grand fir (*A. grandis*) and balsam fir (*A. balsamea*) [16,29], Norway spruce (*Picea abies*) and Sitka spruce (*P. sitchensis*) [24,30], loblolly pine (*P. taeda*), lodgepole pine (*P. contorta*), and jack pine (*P. banksiana*) [22,31]. All the three remaining putative DTPS isolated from Calabrian pine, instead, were found to contain only the class-I signature motifs, plus incomplete versions of the class-II one, lacking D residues known to be critical for class-II catalysis [32] either in the middle (Pnl DTPS3) or in the first and last positions (Pnl DTPS2 and Pnl DTPS4). Although representing putative monofunctional DTPSs, the three sequences only showed 33% to 34% protein sequence identity to the conifer monofunctional class II *ent*-copalyl diphosphates synthases and class I *ent*-kaurene synthases involved in GA metabolism (data not shown), suggesting their roles in specialized, as opposed to general, metabolism.

A phylogenetic analysis including the four deduced amino acid sequences from Calabrian pine and all the pine DTPSs identified in the NCBI database (Figure 3), allowed us to locate the isolated predicted proteins in the four phylogenetic groups in which the *Pinus* members of the TPS-d3 clade can be divided [20].

Based on the sequence relatedness with the previously characterized pine DTPSs, it was possible to predict the potential functions of three out of four DTPSs isolated from Calabrian pine. Pnl DTPS1 was found to cluster in group 1 with the other five bi-I/II-DTPSs, showing 98–99% aa sequence identity with the four of them which have been functionally characterized so far, namely Pc DTPS LAS1, Pc DTPS LAS2, Pb DTPS LAS1 and Pt DTPS LAS1. Of these, the three bifunctional DTPSs from *P. banksiana* and *P. contorta* were shown to produce the diterpene alcohol 13-hydroxy-8 (14)-abietene [22]. This unstable allylic alcohol can undergo dehydration, resulting in the formation of abietadiene, neoabietadiene, palustradiene, and levopimaradiene, consistent with the GC–MS results previously obtained for Pt DTPS LAS from *P. taeda* [31]. On the basis of such sequence similarity, Pnl DTPS1 could be predicted to be involved in the synthesis of abietane-type diterpene olefins. Interestingly, however, when aligned with the other group-1 DTPSs (Appendix A), Pnl DTPS1 from Calabrian pine revealed distinctive amino acids substitutions, namely D/G-515, G/E-565, and D/N-632, which could lead to a change in the protein structure and hence in its product(s) profile.

The Pnl DTPS2 was found to be closely related to four mono-I DTPSs belonging to the phylogenetic group 2 (Figure 3), for which Hall et al. [22] observed no biochemical activity. All of these proteins, though very similar among each other (95% to 98% protein sequence identity), show a low identity both with the above five putative bi-I/II DTPSs from the *Pinus* species (64–65%), and with the other identified pine mono-I DTPSs (73–76%) (Appendix A). Although the four mono-DTPS from *P. contorta* and *P. banksiana* contain the class-I signature motif, and their homology modelling [33] predicts that they do possess a conserved γβα-domain folding pattern [22], the presence of unique structural features near their active sites, conserved also in the Pnl DTPS2 from Calabrian pine (Appendix A), could explain their absence of function. In such a respect, it was proposed that, in these group-2 DTPSs, the side chains of F-592, located upstream of the class I motif, and likewise those of F-814 and H-817, can protrude into the active site cavity and may cause a steric hindrance, possibly impeding catalytic activity [22]. It has been therefore speculated that these enzymes may have evolved from functional DTPSs into a trough of no function, from where they may evolve toward new DTPS activities or simply represent dead-end mutations of functional DTPSs [22].

Based on sequence similarity (Figure 3), and diverging from Pnl DTPS1, Pnl DTPS3 and Pnl DTPS4 were predicted to produce pimarane-type olefins, namely pimaradiene, sandaracopimaradiene, and isopimaradiene. In particular, Pnl DTPS3 was found to cluster in the phylogenetic group 3, together with one protein from *P. contorta* (Pc DTPS mISO1) and one from *P. banksiana* (Pb DTPS mISO1) (Figure 3), both of which were found to produce isopimaradiene as the main product, with small amounts of sandaracopimaradiene [22]. The members of such a group, showing 96% to 99% protein sequence identity among each other, were found to be more similar to the mono-I DTPSs from the phylogenetic group 4 (79–80%) than to those of phylogenetic group 2 (74–76%; Appendix A). Additionally, for the group-3 DTPS, as noted above for the group-1 ones, sequence alignment revealed amino acid substitutions exclusively present in the Pnl DTPS3 from Calabrian pine, namely K/N-642, D/N-748, and H/Y-749 (Appendix A), which could lead to a change in the protein structure and hence in its product(s) profile. Likewise, Pnl DTPS4 was found to cluster in the phylogenetic group 4 (Figure 3), together with two previously described mono-I DTPS, one from *P. banksiana* (Pb DTPS mPIM1) and one from *P. contorta* (Pc DTPS mPIM1), both of which were functionally characterized as forming pimaradiene as their major product [22]. Despite the pronounced sequence identity among the group-4 predicted proteins (about 94%; Appendix A), the high number of amino acid substitutions found in the Pnl DTPS4, compared to the other two DTPSs (Appendix A), suggests that only its functional characterization might elucidate its specific catalytic competence.

Although we tried to predict the potential functions of Calabrian pine DTPSs based on sequence relatedness, it has to be mentioned that examples of an apparent lack of structure–function correlation have been observed in the plants’ TPS family. Hall et al. [34], for instance, reported that conifer monoterpene synthases sharing 80–90% aa identity among each other can catalyse biochemically distinct reactions, while, vice versa, others sharing only 50–60% protein identity among each other can form the same product. For this reason, a functional characterization consisting of heterologous expression in bacterial systems and testing of the recombinant enzymes with their potential terpenoids substrates would be essential to elucidate the actual functions of Calabrian pine DTPSs.

### 2.4. Genomic Organization of Diterpene Synthases in Calabrian Pine on the Background of DTPS Functional Evolution

The genomic sequences encompassing the ORFs of the four *Pnl DTPS*1–4 genes isolated in the present study are schematically shown in Appendix A. These genomic sequences have been deposited in the GeneBank database under the accession numbers OK245422 to OK245425.

The alignment of each genomic sequence with its corresponding cDNA revealed an almost perfect matching among the latter and the exonic regions of the former, thus allowing a reliable determination the exon/intron structure of each DTPS gene. *Pnl DTPS1* and *Pnl DTPS2* were found to contain 16 exons and 15 introns, whereas 15 exons and 14 introns were found in the *Pnl DTPS3* and *Pnl DTPS4* sequences (Appendix A). Apart from the 5′ end, which showed considerable variability in terms of gene structure and sequences, the four *DTPS* genes from Calabrian pine were found to exhibit a high level of conservation of their genomic structural features, in terms of intron location, exon number and size, and position of the class-I active site functional motif (Appendix A). Obvious patterns of intron sizes and sequences were not detected, although there was a strong conservation of their position along the genomic sequences (introns IV to XV in *Pnl DTPS1* and *Pnl DTPS2* and introns III to XIV in *Pnl DTPS3* and *Pnl DTPS4*; Appendix A). The intron sizes were found to be generally small (about 50–200 nt), although some large introns (more than 300 nt) were also detected (Appendix A). In addition, these introns were AT rich, with repetitive sequences rich in T (3–10 mers; data not shown). All the four Calabrian pine DTPS genes were found to contain intron–exon junctions, which, with a few exceptions, followed the GT/AG boundary rules (data not shown) [35]. Moreover, the phasing of the intron insertion, defined as the placement of intron before the first, second, or third nucleotide position of the adjacent codon and referred to as phase 0, 1, and 2, respectively [36], appeared to be equally well conserved (Appendix A).

In an attempt to gain insight into the functional evolution of terpene synthases genes in plants, Trapp and Croteau [37] divided them into three classes, namely I, II, and III, which might have evolved sequentially by intron loss mechanisms. According to such classification, the four Calabrian pine DTPS genes isolated in the present study belong to class I, formed mainly by both mono- and bi-DTPS genes containing 12–14 introns, present in both gymnosperms (secondary metabolism) and angiosperms (primary metabolism). Indeed, the aforementioned authors [37] showed a strong conservation of the genomic structure between the genes encoding monofunctional CPS and KS enzymes of angiosperm GA metabolism, on one side, and a gene coding for the bifunctional DTPS abietadiene synthase from *Abies grandis* (AgAS), involved in specialized metabolism, on the other side. This led the above authors to propose that AgAS might be reminiscent of a putative ancestral bifunctional DTPS from which the monofunctional CPS and KS were derived through gene duplication and the subsequent specialization of each of the duplicated genes for only one of the two ancestral activities. This model of an ancestral bifunctional DTPS was validated later on by the discovery of a bifunctional CPS/KS from the moss model species *Physcomitrella patens*, showing a similarly conserved gene structure [38].

In the present work, the isolation of the complete genomic sequences of Calabrian pine DTPSs made it possible to further and complete the analysis of Trapp and Croteau [37] by comparing them with the DTPSs already assigned to class I (Figure 4). Such comparison confirms that, as already noticed among the four DTPSs from Calabrian pine (see above), number, position, and phase of the introns III-XIV are highly conserved in all the class-I DTPS genes, among which *AgAS*, regarded as descending from a putative ancestral bifunctional DTPS gene (see above). In contrast, number, placement and phase of introns preceding intron III on the 5′terminus side were not conserved among the compared DTPS genes, and an additional, equally not conserved, intron was also found in this region in the genomic sequences of *Pnl DTPS1* and *Pnl DTPS2* (Figure 4).

Even though conifer bifunctional DTPSs of specialized metabolism and monofunctional DTPSs of specialized metabolism and GA biosynthesis represent three separate branches of DTPS evolution [20,22], their conserved gene structure provides strong evidence for a common ancestry of DTPS with general and specialized metabolisms. In agreement with the phylogenetic analysis (Figure 3), the highly conserved genomic organization detected among the four Calabrian pine genes confirmed also that the monofunctional class-I DTPSs of specialized metabolism in *Pinus* species have evolved in relatively recent times by gene duplication of a bifunctional class-I/II DTPS, accompanied by loss of the class-II activity and subsequent functional diversification. It is worth noting that while the bifunctional class-I/II DPTS of Calabrian pine, and the putative homologous proteins from *P. taeda*, *P. contorta* and *P. banksiana* have orthologs in other conifers, e.g., in *P. abies*, *P. sitchensis*, *Abies balsamea* and *A. grandis*, class-I DTPSs of specialized metabolism have not yet been discovered in other conifers outside of the *Pinus* genus. It is therefore conceivable that they constitute a lineage-specific clade of the TPS-d3 group arising from a common ancestor of the closely related species of Calabrian pine, *P. contorta* and *P. banksiana*, and possibly of all the *Pinus* species; after that pine, spruce, and fir genera became separated from each other.

### 2.5. Transcripts Profiling of Calabrian Pine DTPS Genes Reveal Differential Expression across Different Tissues and Suggest Their Putative Roles in the Biosynthesis of Diterpene Resin Acids

The four DTPS genes isolated from Calabrian pine were found to be constitutively expressed in all the five tissues analysed, although their transcription levels were highly variable (Figure 5).

Compared to the other three DTPS genes, *Pnl DTPS1* was highly expressed in LS and IS. The expression levels of such gene were also comparatively high in R, with respect to the very low number of transcripts detected in YN and MN (Figure 5). Overall, the expression pattern of *Pnl DTPS1* in each of the different Calabrian pine tissues was consistent with the corresponding diterpenoids profiles: first, by comparing Figure 5 and Figure 1A, it can be seen that *Pnl DTPS1* transcript abundances and the total amounts of DRAs in the different tissues are essentially correlated. Secondly, by considering its high predicted protein sequence identity with other bi-I/II DTPS from *Pinus* spp. known to produce abietane-type diterpene olefins, namely abietadiene, neoabietadiene, palustradiene, and levopimaradiene (see Section 2.3, above), the expression levels of *Pnl DTPS1* were comparatively higher in those same tissues, namely LS, IS and R, in which abietic and palustric acids were found to be amongst the predominant DRAs (see Figure 1D–F).

Although significantly lower than those of the other two DTPS genes, the expression levels of *Pnl DTPS3* and *Pnl DTPS4* were similar in LS, IS and R, with a comparatively lower amount detected in YN and MN (Figure 5). Again, tissue-specific gene expression levels were found to be consistent with the corresponding DRA profiles: indeed, the predicted protein sequences of Pnl DTPS3 and Pnl DTPS4 were found to be highly homologous with the ISO and the PIM DTPSs from *P. contorta* and *P. banksiana*, respectively (see Section 2.3 above), known to produce pimarane-type olefins, namely pimaradiene, sandaracopimaradiene and isopimaradiene, acting as precursors of the corresponding DRAs. As a matter of fact, pimarane-type DRAs were found to accumulate in considerably lower amounts than the abietane-type DRAs in most of the tested Calabrian pine tissues (see Figure 1D–E).

Among the pimarane-type DRAs, isopimaric acid was significantly more abundant than pimaric acid in most of the tissues tested (Figure 1), although no significant differences were detected in the number of transcripts of the two genes potentially involved in their synthesis, namely *Pnl DTPS3* and *Pnl DTPS4*, respectively (Figure 5, see above). These findings suggest that other TPSs might be involved in the production of isopimaric acid in Calabrian pine. Indeed, bifunctional enzymes producing isopimaric acid have been previously identified from *P. abies* [30], *P. sitchensis* [24], and *A. balsamea* [16], although no obvious ISO candidate has been identified so far in the *Pinus* species [2,34]. It would be conceivable that an orthologous bifunctional ISO enzyme is present in Calabrian pine, which would account for the discrepancy between the transcript abundances and metabolite levels in the analysed tissues.

Finally, transcript levels of *Pnl DTPS2* were the highest in LS and IS, although significantly lower than those of *Pnl DTPS1*, and moderate in R, MN, and YN (Figure 5). It is worth noting that in both types of needles, the expression levels of *Pnl DTPS2* were remarkably higher than those of the other three genes (Figure 5). Because previous attempts to functionally characterize orthologous genes in other pine species were unsuccessful (see Section 2.3 above), it is not possible at present to make correlative hypotheses on the possible role of *Pnl DTPS2* in DRAs biosynthesis. Nonetheless, its sustained and tissue-specific expression levels observed here, which appears to be correlated with the accumulation of dehydroabietic acid (compare Figure 1 and Figure 5), warrant further and deeper studies to elucidate the true function of *Pnl DTPS2* and orthologous genes from *Pinus* species in conifer DRA biosynthesis.

In summary, the diterpenoid profiles determined in the different tissues of Calabrian pine appear to be consistent with the potential roles of three of the four DTPS genes isolated in the present study. It should be noted, however, that none of the DTPS genes isolated here can be associated with the synthesis of dehydroabietic acid, despite the fact that this was one of the most abundant DRAs detected across all the Calabrian pine tissues (Figure 1C–F). As a matter of fact, the biosynthesis of dehydroabietadiene has not been resolved yet in any plant species [22], while one member of the CP450 family in *P. sitchensis* (PsCYP720B4) was found to be able to interact with the dehydroabietadienate group of substrates (dehydroabietadiene, dehydroabietadienol, and dehydroabietadienal) to produce dehydroabietic acid [17].

## 3. Materials and Methods

### 3.1. Plant Material

Three-year old Calabrian pine (*Pinus nigra* subsp. *laricio* (Poiret) Maire) saplings obtained from the Calabria Regional forest nursery (Calabria Verde Agency, Catanzaro, Italy) were grown in the open within protective housings set up at the Calabria Regional Biodiversity Observatory, located at Cucullaro (38° 17′27″ N, 15° 81′68″ E; altitude 1010 MASL, exposed east), in the heart of Aspromonte National Park, southern Italy. In the course of four sampling campaigns from November 2019 to May 2020, different tissues/organs were collected, namely young needles (YN), mature needles (MN), bark and xylem combined from leader stem (LS), bark and xylem combined from interwhorl stem (IS), and roots (R). All collected tissues were immediately frozen in liquid nitrogen and stored at −80 °C until analysis.

### 3.2. Extraction and GC/MS Analysis of Diterpene Metabolites

After thawing, tissue samples were dried (48–72 h in the dark) at room temperature and then cut into fragments of about 1–2 mm by means of a scalpel. For all the tissue types, the extraction of the diterpenoid fraction was performed following the procedure described by López-Goldar et al. [28] with minor modifications. Briefly, approximately 250 mg of each of the five different tissue types were extracted twice with 2 mL of a n-hexane/dichloromethane mixture (1:1; *v*/*v*). During each extraction cycle, the extracts were kept in an ultrasonic bath at 25 °C for 20 min. After pooling together the two aliquots obtained in a recovery glass vial, residual water was removed by passing the extracts onto a column containing 2 g of anhydrous Na_2_SO_4_, and the obtained eluates were kept in the dark and stored at −20 °C. For derivatisation, first 200 μL of each extract were passed onto a column containing 15 mg of graphitized carbon, to remove non-terpenic impurities, and then 50 µL of each eluate were transferred into a conical vial and dried under a gentle stream of N_2_. After drying, 100 µL of a 1:1 (*v*/*v*) mix of N,O-bis (trimethylsilyl) trifluoroacetamide, containing 1% (*v*/*v*) trimethylchlorosilane, plus pyridine were added to each sample, and the derivatization was allowed to proceed for 30 min at 65 °C. Finally, the solution was brought to dryness under a gentle stream of N_2_, the residue was resuspended with 50 µL of n-hexane and finally stored in darkness at −20 °C until GC-MS analysis. For each of the aforementioned tissue types, three biological replicates were processed and analysed, each of them in triplicate.

Qualitative and quantitative analysis of diterpenes from Calabrian pine tissues were carried out by means of a high–fast GC-MS approach an Agilent Technologies GC (model 7890A, Santa Clara, CA, USA), equipped with a VF-5ms capillary column (Agilent Technologies; 15 m × 0.15 mm of inner diameter and a 0.15 μm film thickness) under the following thermal conditions: from 90 °C (2 min) to 350 °C with a ramp of 44.7 °C min^−1^, then isothermal for 5 min. The He carrier gas constant flow was 1.2 mL min^−1^. The sample injection (0.5 µL) was performed under the pulsed splitless technique (43 psi) at 300 °C. The coupled detector consisted of an Agilent mass selective detector (VL MSD-Triple-Axis Detector), mod. 5975C. The transfer line, the ion source and the analyser were kept at 300 °C, 230 °C and 150 °C, respectively. The acquisition was carried out under full scan mode (range m/z: 50–650).

The identification of the different diterpene metabolites was carried out by comparison of experimental mass spectra both with those in NIST08 and Wiley02 Libraries and those of the available reference literature [22,31,39], as well as of their related retention indices [28]. As far as the Wiley and NIST mass spectra libraries are concerned, the spectral match scores obtained for the diterpenes analysed in the present work were invariably higher than 850, consistently returning the correct identification of each metabolite as the “first hit”. According to the NIST library guidelines, the above score value of mass spectra match is considered to be satisfactory and reliable for the correct identification of a given molecules.

The analyte concentrations, expressed as µg g^−1^ dry weight (d.w.), were calculated by comparison with a calibration curve obtained by using a commercial standard of abietic acid (1R,4aR,4bR,10aR)-1,4a-dimethyl-7-(propan-2-yl)-1,2,3,4,4a,4b,5,6,10,10a-decahydrophenanthrene-1-carboxylic acid (Sigma-Aldrich catalog N. 00010).

The GC/MS methods used in the present study for the extraction and analysis of plant metabolites were adequately validated for their selectivity, precision, and efficiency. Selectivity was verified by observing that no interfering peak was apparent at the elution time of each target analyte upon injecting three replicate blank samples. Precision was tested by measuring the inter- and intra-day variability in the chromatographic profiles of spiked samples, which ranged from 2 to 7% in terms of relative standard deviation. Finally, the intrinsic recovery of the extraction method was calculated as a mean of three replicate samples, in each of which the plant tissue was spiked with a known aliquot of abietic acid standard solution and then extracted, cleaned, and derivatized prior to injection onto GC-MS. Regardless of the tissue extracted, the measured mean recovery always ranged from 80 to 90%.

### 3.3. RNA Isolation and cDNA Synthesis

Total RNA was extracted from 250 mg of each of the five tissues considered according to Pavy et al. [40]. RNA concentration and integrity were checked using a NanoDrop ND-1000 spectrophotometer (Labtech, East Sussex, UK). Only RNA samples with a 260/280 wavelength ratio between 1.9 and 2.1, and a 260/230 wavelength ratio greater than 2.0, were used for cDNA synthesis. First-strand cDNA was synthesized from 3 μg of total RNA of each of the five tissues using a Xpert cDNA Synthesis Kit (GRiSP Research Solution, Porto, Portugal) according to the manufacturer’s instructions.

### 3.4. DNA Extraction

Genomic DNA was extracted from 100 mg of young and mature needles using a NucleoSpin^®^ Plant II kit (Macherey-Nagel, Düren, Germany) according to the manufacturer’s instructions. The integrity and concentration of DNA were determined by 0.8 (*w*/*v*) agarose gels stained with ethidium bromide (0.001%) using known concentrations of unrestricted lambda DNA as control.

### 3.5. Isolation of Partial and Full-Length cDNAs Coding for Diterpene Synthases

According to the methods reported in Alicandri et al. [20], RT-polymerase chain reaction (PCR) was used to amplify partial cDNA coding for DTPSs in *P. nigra* subsp. *laricio* by using forward and reverse primers designed in conserved regions among DTPS sequences of *Pinus* species of the different groups identified by phylogenetic analysis. The complete list of the used forward and reverse primers is reported in Appendix A.

Each PCR reaction was performed in a total volume of 50 μL containing 2 μL of RT reaction obtained from a pool of total RNA from the five different tissues (see Section 3.3), 0.4 μM of each forward and reverse primer, and 25 μL of Xpert Taq Mastermix (2X) (GRiSP Research Solutions, Porto, Portugal), which includes pure Xpert Taq DNA Polymerase, dNTPs, MgCl_2_ and optimized PCR buffer.

All reactions were carried out in an Eppendorf Thermal Cycler (Master cycler Gradient) with the following parameters: initial denaturation at 95 °C for 5 min, 35 cycles of amplification, each at 95 °C for 1 min, 58–62 °C (depending on the annealing temperature of the primers) for 1 min, 72 °C for 3 min, and a final extension at 72 °C for 5 min.

The partial DTPS cDNAs were used as templates for 5′ and 3′ RACE extensions using the 5′/3′ RACE System for Rapid Amplification of cDNA Ends Kit from INVITROGEN Life Technologies, following the manufacturer’s instructions and using 3 μg of a pool of total RNA from the five different tissues. The sequences of the RACE primers used are reported in Appendix A.

### 3.6. Isolation of Genomic Sequences Coding for Diterpene Synthases

Genomic DNA was used to amplify *P.nigra* subsp. *laricio* DTPS genomic sequences by using specific forward and reverse primers designed, respectively, on the proximity of the initiation (ATG) and on the stop codons of each full-length isolated cDNA (Appendix A). The PCR reactions and conditions were the same as described in Section 3.5 [20], with the exception of the extension step that was increased from 3 to 6 min at 72 °C.

### 3.7. Cloning and Sequencing of RACE, cDNA and Genomic Amplification Products

Samples (5–10 μL) of the amplification products of RACE, partial cDNAs and genomic sequences were separated on 1.5% agarose gels and visualized under UV radiation after staining with ethidium bromide (0.001% *w*/*v*) by using the UVITEC Essential V6 Gel Imaging and Documentation System (Cleaver Scientific, Rugby, United Kingdom). PCR products of expected size were excised from the gel, purified using the High Pure Purification kit (Roche, Mannheim, Germany) according to the manufacturer’s instructions, and cloned into the pGEM-T easy plasmid vector (Promega, Madison, WI, USA) following the manufacturer’s protocol. Three different clones for each cDNA, genomic and RACE amplicon were sequenced. Plasmid DNA for a sequencing reaction was prepared from 3 mL overnight cultures using a Wizard^®^ Plus SV Minipreps DNA Purification Systems (Promega, Madison, WI, USA). A private company (MWG, Biotech AG, Germany) performed sequencing. Recombinant positive plasmids were sequenced on both strands by the ABI PRISM 377 capillary sequencer (PE Applied Biosystem, Waltham, MA, United States) using an ABI Prism Dye Terminator sequencing kit (PE Applied Biosystem) and either vector or sequence specific primers. The sequences of the genomic clones were obtained by sequencing them with internal primers complementary to the cDNA sequences, and designed near the predicted exon/intron junctions so as to amplify each exon and nearby intron on both strands to fill gaps and resolve uncertainties (primers are available upon request).

### 3.8. Analysis of the Nucleotide and of the Deduced Amino Acid Sequences

All the nucleotide sequences obtained were analysed by DNAMAN Sequence Analysis Software (Version 3, Lynnon Biosoft) and their homologies were scored using the BLASTX program through the National Center for Biotechnology Information (NCBI) database. The software developed by NetGene [41] was used for the prediction of intron splice sites within the genomic sequences. The predicted protein sequences were analysed by searching for conserved motifs in CDD (Conserved Domain Database in the NCBI) and SMART (Simple Modular Architecture Research Tool, European Molecular Biology Laboratory) databases; their subcellular locations were predicted by ChloroP [42], Predotar [43], and WoLF PSORT [44].

### 3.9. Phylogenetic Analysis

A multiple-sequence alignment of pine DTPS deduced proteins was performed by ClustalX version 1.83 [45], using the Gonnet series as the protein weight matrix and parameters set to 10 gap open penalty, 0.2 gap extension penalty, negative matrix on, and divergent sequences delay at 30%. The *ent*-kaurene synthase from *Physcomitrella patens* (BAF61135) was also included in the analysis as outgroup. A phylogenetic tree was generated with the Neighbor-Joining method [46] using MEGA X software [47]. The evolutionary distances were computed using the JTT matrix-based method and are in the units of the number of amino acid substitutions per site. The rate variation among sites was modeled with a gamma distribution (shape parameter = 1). The reliability of the tree obtained was tested using bootstrapping with 1000 replicates.

### 3.10. Gene Expression Analysis

The expression patterns of the isolated *P. nigra* subsp. *laricio* DTPS sequences were analysed in the five tissue types considered by quantitative real time (qRT-PCR).

As for the reference genes for expression analysis, we looked at those showing stable expression in different pine tissues in the presence of stress conditions of different origin [48,49]. The reference genes selected encode the following proteins: Actin 1 (*ACT1*, NCBI accession no KM496527), Cyclophilin (*CYP*, KM496534), Tubulin alpha (*TUBα*, KM496535), Polyubiquitin 4 (*UBI4*, KM496539), and uncharacterized protein LOC103705956 (*upLOC*, MN172175).

Quantitative RT-PCR analysis was performed using the AriaMX real-time PCR system with the Fast Q-PCR Master Mix (SMOBIO, Hsinchu, Taiwan) according to the manufacturer’s protocol. Each reaction was run in a 20 μL final volume containing 1 μL of cDNA, and 150 nM forward and reverse primers. No template and RT-minus controls were run to detect contamination, dimer formation, or the presence of genomic DNA. Specific primer pairs were designed both for the target and the selected reference genes using the Beacon Designer 6 software (Stratagene, La Jolla, CA), and the following stringency criteria: Tm of 55 °C ± 2 °C; PCR amplicon length between 60 and 200 bp; primer length of 21 ± 3 nt; and 40% to 60% guanine-cytosine content. Primers were also designed at the 3′ end of each sequence, to encompass all potential splice variants and ensure equal RT efficiencies. Only primer pairs generating a sharp peak by melting curve analysis (without unspecific products or primer–dimer artifacts) and showing efficiencies between 90 and 110%, and R^2^ values (coefficient of determination) calculated for standard curves higher than 0.995, were selected for expression analysis of the target and references genes.

Standard curves based on five points, corresponding to a five-fold dilution series (1:1–1:243) from pooled cDNA, were used to compute the PCR efficiency of each primer pair. The PCR efficiency (E) was derived by the eq. E = (10[−1/m] − 1) × 100, where m is the slope of the linear regression model fitted over log-transformed data of the input cDNA concentration versus Ct values, according to the linear equation y = m × log(x) + b. The thermal profile comprised three segments: 95 °C for 2 min, 40 cycles of 15 s denaturation at 95 °C, 1 min annealing at 56 °C and the dissociation curve, consisting of 1 min incubation at 95 °C, 30 s incubation at 60 °C and a ramp up to 95 °C. Three biological replicates, resulting from three different RNA extractions, were used in the quantification analysis. Three technical replicates were analysed for each biological replicate.

Raw Ct values were transformed to relative quantities by using the delta-Ct formula Q = E^ΔCt^, where E is the efficiency of the primer pair used in the amplification of a specific gene (100% = 2), and ΔCt is the difference between the sample with the lowest Ct (highest expression) from the dataset and the Ct value of the sample in question. In particular, for the comparison of the relative expression levels of the isolated DTPSs in all tissue types the formula used to convert Ct values into relative quantities was Q = 2^ΔCt^. This assumption was justified by the fact that the amplification efficiencies of the considered genes were approximately the same, ranging from 96 to 100%.

The expression stability of the aforementioned candidate reference genes was evaluated by the software program NormFinder [50], as described in Paolacci et al. [51]. The best two-gene combination proposed by NormFinder was *CYP* + *upLOC*, with a stability score considerably lower than that of the most stable single reference gene, namely *CYP*, among those considered. Therefore, the expression levels of the genes of interest were normalized against the geometric mean of the two aforementioned reference genes, and their normalized relative values reported as mean value ± SD. Standard deviation values for normalized expression levels were calculated according to the geNorm user manual (geNorm manual, updated 8 July 2008).

### 3.11. Statistical Analysis

Each reported value for metabolites and gene expression levels represents the mean of a total of nine values, obtained from three biological replicates and three technical replicates for each biological replicate. The statistical significance of the differences was evaluated by one-way ANOVA followed by the Tukey’s test.

### 3.12. Ethical Statement

Voucher specimens of the collected plant material have been deposited in the Department of Agriculture of the *Mediterranean* University of Reggio Calabria, Italy. Depending on the tissue considered, total amounts ranging from 500 to 2500 g were collected during the sampling campaigns.

## 4. Conclusions

In the present study, we carried out for the first time, to the best of our knowledge, a quali-quantitative analysis of diterpenoid composition in tissues obtained from different organs of *Pinus nigra* subsp. *laricio* (Poiret) Maire (Calabrian pine), namely young and mature needles, leader stem, interwhorl stem, and roots. In these same tissues, we carried out the isolation and sequencing of full-length cDNAs and of the corresponding genomic sequences encoding for diterpene synthases involved in the specialized diterpenoid metabolism.

It was shown that diterpene resin acids are the most abundant diterpenoids across all the examined tissue types, together with remarkably lower amounts of the corresponding aldehydes and olefins. All the Calabrian pine tissues examined showed the presence of the same nine diterpene resin acids, whose quantitative distribution was nonetheless found to be remarkably different in the different tissues examined. In agreement with other studies on the model *Pinus* species, our analyses showed that abietane-type DRAs were more abundant than pimarane-type DRAs in all the Calabrian pine tissues, even though there were considerable differences in their amounts among the different tissues. Taken together, the above results seem to suggest a remarkable tissue specificity, as well as species specificity, of terpenoid composition in conifers, whose functional significance in terms of the plant’s biological performance, as well as in terms of possible exploitation for a variety of applications, awaits and deserves further study.

A phylogeny-based approach allowed the isolation and sequencing of four partial and full length cDNAs coding for diterpene synthases in Calabrian pine, denoted as *Pnl DTPS1*, *Pnl DTPS2*, *Pnl DTPS3*, and *Pnl DTPS4*, with each of the corresponding encoded proteins found to belong to one of the four groups into which the d3 clade of the plants’ terpene synthase family can be divided. The subsequent analysis of the deduced amino acid sequences allowed us to predict that both monofunctional, such as Pnl DTPS2-4, and bifunctional, such as Pnl DTPS1, diterpene synthases are involved in the biosynthesis of diterpene resin acids in Calabrian pine. Transcript profiling of the Calabrian pine DTPS genes revealed differential expression across the different tissues and were found to be consistent with the corresponding diterpenoids profiles, suggesting potential roles for three of the four DTPSs genes in the biosynthesis of diterpene resin acids.

Finally, the obtained full-length DTPS cDNAs were also used to isolate the corresponding complete genomic sequences, for each of which the exon/intron structure was determined. This allowed us to place the DTPS genes isolated from Calabrian pine into the background of the current ideas on the functional evolution of diterpene synthases in plants and, in particular, on the functional diversification accompanying genera and species evolutionary segregation within the gymnosperms.

Beyond their roles in conifer defence, because of their ample physical and chemical diversity and their resulting technological versatility, diterpene resin acids provide a large-volume, renewable resource for industrial and pharmaceutical bioproducts. Therefore, novel and in-depth knowledge of the evolutionary diversification of members of the conifer DTPS family, their modular structure, and their putative functions appears to be important not only for a deeper understanding of their physiological and ecological roles, but also to foster metabolic engineering and synthetic biology tools for the production of high-value terpenoid compounds.

## Figures and Tables

**Figure 1 plants-10-02391-f001:**
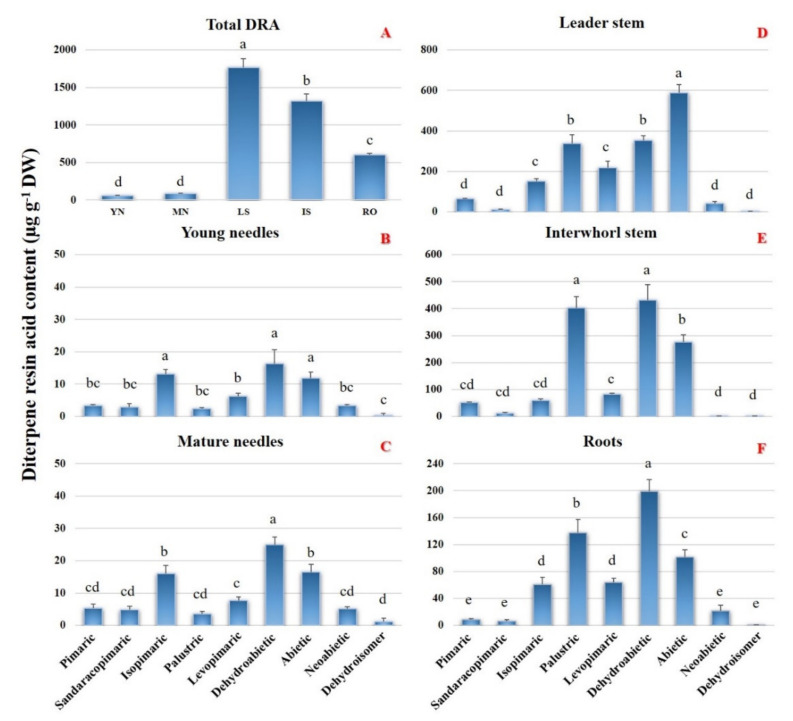
Total diterpene resin acids (DRAs, panel at the top) and levels of individual DRAs in different tissues of 3-year-old *Pinus nigra* subsp. *laricio* (Calabrian pine) saplings. Error bars indicate the standard deviation of the mean. The statistical significance of the differences was evaluated by one-way ANOVA, followed by the Tukey’s test. Different letters denote statistical significance of the difference at *p* < 0.01. DW, dry weight.

**Figure 2 plants-10-02391-f002:**
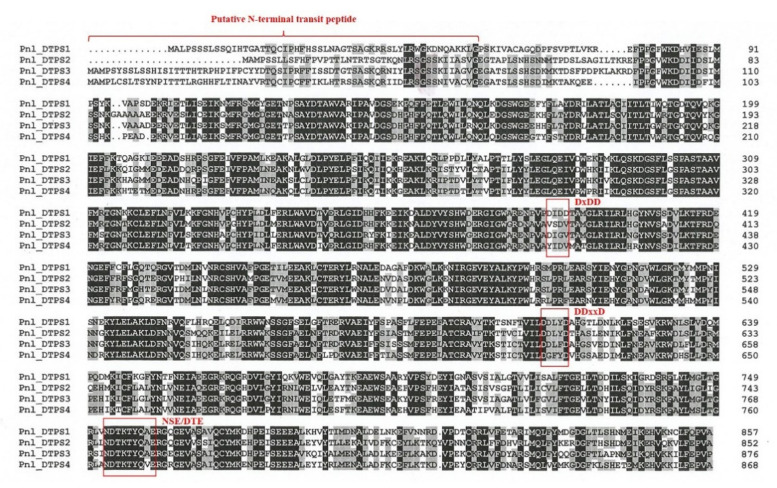
Alignment of deduced amino acid sequences of the four putative diterpene synthases from Calabrian pine (Pnl DTPS1–4) isolated in the present study. Amino acid residues with black background indicate highly conserved regions, while amino acid residues which are identical in more than 50% of the proteins are in grey background. The DTPS class II (DxDD) and class I (DDxxD, NSE/DTE) signature motifs are indicated.

**Figure 3 plants-10-02391-f003:**
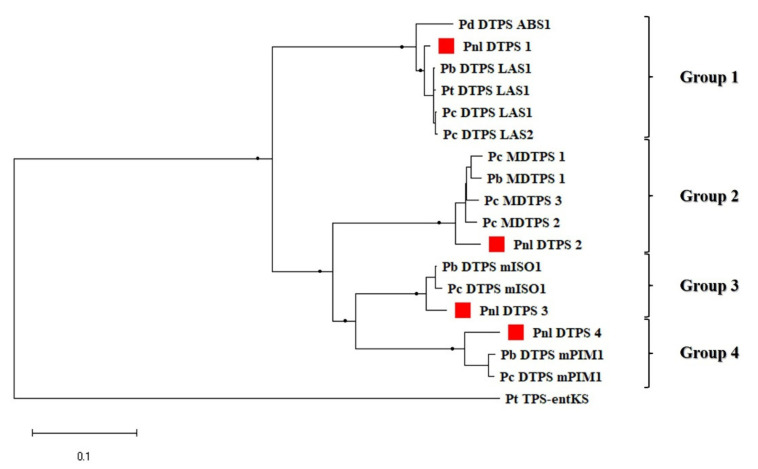
Phylogenetic tree of the deduced amino acid sequences of 13 diterpene synthases (DTPSs) identified in different *Pinus* species (Appendix A) and the four DTPSs from Calabrian pine isolated in the present study (red squares). The *ent*-kaurene synthase from *Physcomitrella patens* (Pt TPS-entKS, BAF61135) was used to root the tree. Branches marked with dots represent bootstrap support more than 80% (1000 repetitions). The four phylogenetic groups identified in the pine members of the d3 clade of terpene synthases are indicated by square brackets.

**Figure 4 plants-10-02391-f004:**
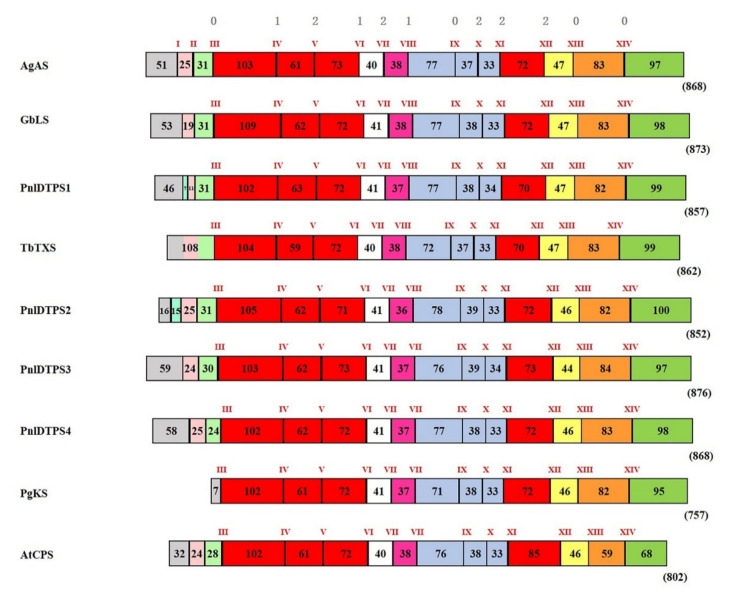
Genomic organization of plant diterpene synthase (DTPS) genes. Black vertical slashes represent introns (indicated by Roman numerals) and are separated among each other by colored boxes with indicated lengths in amino acids, representing exons. The numbers above the introns of the first row from the top represent the intron phase type classification according to Li [33] and indicate conservation throughout the plants’ DTPS genes. Schematization, intron number, and exon coloring scheme are based upon Trapp and Croteau [31]. Genomic DNA sequences compared are as follows: *AgAS*, *Abies grandis* abietadiene synthase (NCBI accession no. AF326516); *GbLS*, *Ginkgo biloba* levopimaradiene synthase (AY574248); *TbTXS*, *Taxus brevifolia* taxadiene synthase (AF326519); *PgKS*, *Pinus glauca ent*-kaurene synthase (GU059905); *AtCPS*, *Arabidopsis thaliana* copalyl diphosphate synthase (AT4G02780); *Pnl DTPS*1–4 denote the DTPS isolated from Calabrian pine in the present study.

**Figure 5 plants-10-02391-f005:**
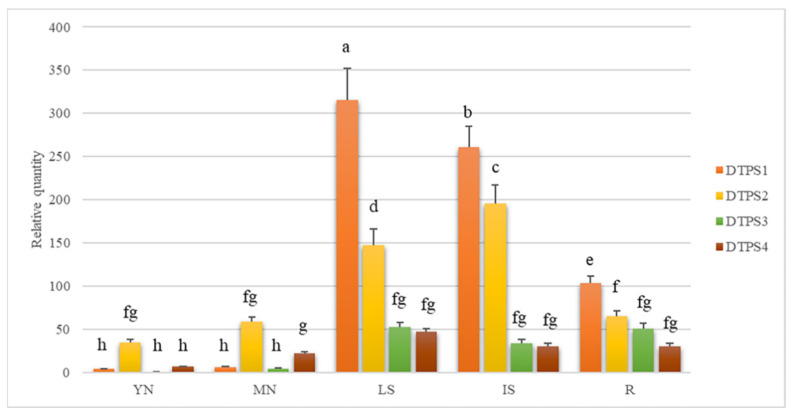
Relative expression levels of four diterpene synthase genes (DTPS1–4) in five different tissues of Calabrian pine. The expression data of each gene were normalized using the geometric average of the two reference genes *CYP* and *upLOC*. Relative expression levels of the different DTPS genes were referred to a calibrator, set to the value 1, which was represented by the gene in the five tissues with the lowest expression (*DTPS3* in YN). YN, young needles; MN, mature needles; LS, bark and xylem combined from the leader stem; IS, bark and xylem combined from the interwhorl stems; R, roots. Different letters denote significant differences according to the Tukey’s test (*p* < 0.01).

## Data Availability

The data contained within the present article and in its Appendix A are freely available upon request to the corresponding author.

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
