# Peer review of "Diterpene Resin Acids and Olefins in Calabrian Pine (Pinus nigra subsp. laricio (Poiret) Maire) Oleoresin: GC-MS Profiling of Major Diterpenoids in Different Plant Organs, Molecular Identification and Expression Analysis of Diterpene Synthase Genes"

_plants, 2021, doi:10.3390/plants10112391_

Round 1

Reviewer 1 Report

Overall well written with only minor errors and report interesting findings. There are a few points regarding the GC-MS analysis the authors might address and/or clarify (See attachment).

Author Response

Response to Reviewer 1 Comments

Point 1: Analysis of the analytes of interest involve extraction, cleanup and derivatization steps. Has the analytical method been validated for analyte recovery and precision of recovery to ensure accuracy of results? I tried to follow up the cited references but could not find method validation data. If there is, the authors might want to cite it.

Response 1: The GC/MS methods used in the present study for the extraction and analysis of plant metabolites were adequately validated for their selectivity, precision, and efficiency. Selectivity was verified by observing that no interfering peak was apparent at the elution time of each target analyte upon injecting three replicate blank samples. Precision was tested by measuring the inter- and intra-day variability in the chromatographic profiles of spiked samples, which ranged from 2 to 7% in terms of relative standard deviation. Finally, the intrinsic recovery of the extraction method was calculated as a mean of three replicate samples, in each of which the plant tissue was spiked with a known aliquot of abietic acid standard solution and then extracted, cleaned and derivatized prior to injection onto GC-MS. Regardless of the tissue extracted, the measured mean recovery always ranged from 80 to 90%. The above details concerning the GC/MS analytical methods have been inserted in the present revised version at lines 573-583.

Point 2: From my reading of the paper, a single standard, abietic acid, was used to quantify all the other analytes. This of course assumes the detector response is the same for all the analytes. The authors could comment on the degree of uncertainty introduced by this assumption.

Response 2: The metabolites analysed in the present study are all isomers, with a low degree of variability around a basic and common chemical structure, except for the dehydro-derivatives, having a slightly lower molecular mass due to the presence of an aromatic ring in their structures. On such basis, and as suggested by the dedicated literature, the response factors of the metabolites measured here are usually assumed to be comparable among each other. Therefore, as a first approximation, the degree of uncertainty of the analytical data was incorporated into the estimated value of analytical precision.

Point 3: Chromatographic peak identity was achieved by comparison with reference libraries. These comparisons usually include a % match to indicate how good the match is. The authors could indicate a range and average % match.

Response 3: As far as the Wiley and NIST mass spectra libraries are concerned, the spectral match scores obtained for the diterpenes analysed in the present work were invariably higher than 850, consistently returning the correct identification of each metabolite as the “first hit”. According to the NIST library guidelines, the above score value of mass spectra match is considered to be satisfactory and reliable for the correct identification of a given molecules. The above details concerning the spectral match scores have been inserted in the present revised version at lines 563-568.

Reviewer 2 Report

Authors describe a quali-quantitative analysis of diterpenoids composition in tissues obtained from different organs of the Calabrina pine (Pinus nigra subsp. laricio). Young and mature needles, leader stem, interwhorl stem, and roots were extracted and the diterpen profiles (olefins and acids) were described, mainly detailing the nine commonly observed diterpenic aromatic acids from pimarane, abietane and dehydroabietane skeletons, with a more abundant presence of abietane-type acids, as expected to Pinus especies. Since the huge ecological importance observed to these diterpenes in conifers, the observed tissue-specificity and species-specificity diterpenoid chemical composition deserves further studies. The characteristic chemical/chromatografic profiles observed by GC-MS after derivatization of the acids were supported by transcripts profiling of the DTPSs genes, that revealed differential expression across the different tissues.
Figure 1 could be improved to allow better comprehension of the data. Is is not clear why olefins were presented only briefely on lines 180-185, not pointed out at the chromatogram nor discussed quantitavely in terms of genetic expression of diterpenoids.

Some minor adjustments should be performed, such as low caps initiating sentences, reference format and lack of references at sections 3.5, 3.6 and 3.7.

Author Response

Response to Reviewer 2 Comments

Point 1: Figure 1 could be improved to allow better comprehension of the data.

Response 1: Following the suggestion from the Reviewer 2, Figure 1 was graphically improved and inserted on page 5 of the present revised version.  In our opinion, the modified horizontal layout and the greater font size of its x/y axes labels should have adequately improved the readability of the figure.

Point 2: Is not clear why olefins were presented only briefly on lines 180-185, not pointed out at the chromatogram nor discussed quantitavely in terms of genetic expression of diterpenoids.

Response 2: A representative full scan GC-MS chromatogram (m/z 50-650) of olefins and aldehydic compounds from Calabrian pine tissues was added to the Supplementary Materials as Figure S5, and mentioned as such at line 189 of the present revised version. As highlighted already in the original submission, the estimated concentrations of the olefins and of their related aldehydic forms were several orders of magnitude below those of their corresponding diterpene resin acids (DRAs), and showed a very low signal-to-noise ratio (S/N), in the range 3-10. Because such very low S/N values are usually considered to be analytically meaningful only for the qualitative interpretation of the data, olefins have been described here only in qualitative terms, to complete the description of the chromatographically identified species. Instead, and for these same reasons, we focused exclusively on DRAs our analysis of the quantitative relationships among genes expression and accumulation of diterpenoids, being the DRAs by far the most abundant terpenoids across all the examined tissue types.

Point 3: Some minor adjustments should be performed.

3a) low caps initiating sentences.

Response 3a: After a detailed check of all the main text of the manuscript, no low caps initiating sentences were found.

3b) reference format.

Response 3b: Changes have been made in the present revised version at lines 876, 880, 933-934 and 985.

3c) lack of references at sections 3.5, 3.6 and 3.7.

Response 3c: References have been added to sections 3.5 and 3.6, at lines 600 and 624, respectively.  As for section 3.7, the methods employed were either directly taken from the protocols accompanying the commercial kits and reagents used, or patented by the providers of external services, as in the case of nucleic acids sequencing.

Reviewer 3 Report

In this study, authors evaluated in five different Calabrian pine tissues, namely young- (YN) and mature (MN) needles, bark and xylem combined from leader- (LS) and interwhorl (IS) stems, and roots (R). And isolate four genes corresponded to diterpene synthase genes putatively. However, there is no any function identification for these four gene. Sometime just only few amino acid residues changed cause different products in terpene synthase gene. Thus, it’s no sense to discuss the amino acid composition without function identification of genes. In addition, evolution diversity has reported previously. In this study, genome analysis should discuss other conifer species to evaluate the evolution. Finally, the gene expression didn’t match the GCMS profile. The relationship among products also need to describe in this study.    

Author Response

Response to Reviewer 3 Comments

Point 1: In this study, authors evaluated in five different Calabrian pine tissues, namely young- (YN) and mature (MN) needles, bark and xylem combined from leader- (LS) and interwhorl (IS) stems, and roots (R). And isolate four genes corresponded to diterpene synthase genes putatively. However, there is no any function identification for these four gene. Sometime just only few amino acid residues changed cause different products in terpene synthase gene. Thus, it’s no sense to discuss the amino acid composition without function identification of genes.

Response 1: We are aware that the functional analysis of the full-length transcripts coding for DTPS by their heterologous expression in bacterial or yeast systems and in vitro enzyme assays would be crucial to elucidate the actual functions of Calabrian pine DTPSs, as also recognized at the end of the section 2.3 (lines 334-342 of the original submission). In this context, we are indeed planning to functionally characterize not only the complete transcripts encoding for the diterpene synthases but also those coding for the monoterpene synthases, that in the meantime we have already isolated and about to submit in a manuscript in preparation. It is important to remark here, however, that, unlike monoterpene synthases, in the case of diterpene synthases there seems to be no ambiguity in assigning the same function to proteins isolated from different Pinus species, providing they show a high level of homology among each other and belong to one of the four phylogenetic groups identified so far. On the basis of the work from Hall and Co-workers (2013), in fact, and apart from the proteins assigned to phylogenetic group 2, for which no biochemical activity was observed, the four proteins from three different Pinus species assigned to the phylogenetic group 1 seem to be all involved in the synthesis of abietane-type diterpene oleofins. Moreover, the two proteins from Pinus banksiana and Pinus contorta belonging to phylogenetic group 3 were found to produce isopimaradiene as the main product, while the two proteins from the same above species assigned to the phylogenetic group 4 were shown to produce pimaradiene. Taken together, and as a preliminary step, the above evidence led us to predict the potential functions of the DTPS proteins from Calabrian pine on the basis of the sequence relatedness of each of them with previously characterized pine DTPSs, taking into account phylogenesis. Such predictions of ours, moreover, were supported by the fact that the transcripts profiling of the Calabrian pine DTPS genes across different tissues was found to be consistent, as a whole, with that of the corresponding diterpenoids profiles (see also Response 3 below), thus confirming the potential roles of the isolated DTPS genes presumed on the basis of their sequence homology.

Point 2: In addition, evolution diversity has reported previously. In this study, genome analysis should discuss other conifer species to evaluate the evolution.

Response 2: As stated in the manuscript, the isolation and characterization of the complete genomic sequences of Calabrian pine DTPSs allowed us to further and complete the analysis put forward by Trapp and Croteau (2001). To this end, and as shown in the Fig 4 of the manuscript, the genomic structures of the four Calabrian pine  DTPS were compared, on one side, with the available genomic sequences of monofunctional and bifunctional DTPS involved in terpenoids (“specialized”) metabolism in gymnosperms,  namely AgAS, GbLS and TbTXS, and, on the other hand, with the genomic sequences of monofunctional DTPS involved in gibberellin (“primary”) metabolism, either in gymnosperms, namely PgKS, or in angiosperms, namely AtCPS. As far as the genomic sequences of specialized DTPS genes are concerned, all the above implies, and Fig. 4 demonstrates, that, before the present study was carried out, the only available benchmarks in conifers were the gene encoding the bifunctional DTPS abietadiene synthase from Abies grandis (AgAS) and the gene encoding the monofunctional DTPS taxadiene synthase from Taxus brevifolia (TbTXS). This, on one side, made it not feasible to evaluate at the genomic level the evolution of specialized DTPSs against the background of several other conifer species, as requested by Reviewer 3, while, on the other side, it highlights the contribution of the present study, in which the genomic sequences of both monofunctional and bifunctional specialized DTPS have been reported for the first time in a species belonging to the Pinaceae.

Point 3: Finally, the gene expression didn’t match the GCMS profile. The relationship among products also need to describe in this study.  

Response 3: As far as the first sentence of this Point is concerned, and on the basis of the results presented and discussed in our manuscript, we are unable to agree with the opinion of the Reviewer 3.

First, the diterpenoid profiles measured in the different tissues of Calabrian pine appeared to be consistent with the presumed roles of three out of four DTPSs isolated in the present study. Indeed, the action of the bifunctional class I/II DTPS, namely Pnl DTPS1, which we predicted to be involved in the synthesis of the abietane-type diterpene olefins (abietadiene, neoabietadiene, palustradiene, and levopimaradiene), and the action of the two putative monofunctional class I DTPSs (Pnl DTPS3 and Pnl DTPS4), which likely form pimarane-type olefins (isopimaradiene, pimaradiene, and sandaracopimaradiene), could explain together the presence of seven out of eight predominant diterpene structures actually detected in the Calabrian pine tissues. As pointed out in the section 2.5 of our manuscript, none of the DTPSs isolated from Calabrian pine could be supposed to be involved in the formation of dehydroabietic acid, although this was one of the most abundant DRA detected across all the tested tissue types. As a matter of fact, however, and as discussed in our manuscript, the biosynthesis of dehydroabietadiene has not been resolved yet in any plant species.

Secondly, the tissue-specific transcript profiling of three out of four Calabrian pine DTPSs was found to be consistent with the tissue-specific accumulation patterns of the different DRAs. Indeed, Pnl DTPS1, encoding for a putative bifunctional class I/II DTPS, showed the highest transcript levels in bark and xylem from leader and interwhorl stems, consistently with the high levels of abietane-type DRAs, in particular abietic and palustric acids, found in these same tissues. Furthermore, in these same two tissue types, the transcription levels of Pnl DTPS3 and Pnl DTPS4, encoding for the two putative monofunctional class-I DTPS possibly involved in the synthesis of pimarane-type olefins, were about from 6- to 9-fold lower than those of Pnl DTPS1, again consistently with the lower levels of the pimarane-type DRAs.  In addition, the expression level of Pnl DTPS1 was significantly higher than those of Pnl DTPS3 and Pnl DTPS4 also in roots, in which, although in lower amounts than in leader and interwhorl stems, the abietane-type DRAs prevailed over the pimarane-type DRAs. Finally, in agreement with the very low accumulation of the total DRAs in young and mature needles, the three aforementioned DTPS genes showed the lowest expression levels across the different tissue tested, with no statistically significant difference among each other. Taken together, the above results lead us to suggest a clear relationship between the abundance of gene transcripts and the accumulations of the different types of DRAs across the tested tissues from Calabrian pine.

As far as the second sentence of the Reviewer’s Point is concerned, we apologize we were unable to understand its meaning, and therefore the nature of the changes/correction requested.

Literature cited in the responses to Reviewer 3:

Hall, D.E.; Zerbe, P.; Jancsik, S.; Quesada, A.L.; Dullat, H.; Madilao, L.L.; Yuen, M.; Bohlmann, J. Evolution of conifer diterpene synthases: diterpene resin acid biosynthesis in lodgepole pine and jack pine involves monofunctional and bifunctional diterpene synthases. Plant Physiol 2013, 161, 600–616. https://doi.org/10.1104/pp.112.208546

Trapp, S.C.; Croteau, R.B. Genomic Organization of plant terpene synthases and molecular evolutionary implications. Genetics 2001, 158, 811–832. https://doi.org/10.1093/genetics/158.2.811

Reviewer 4 Report

See report as pdf attached!

Author Response

Response to Reviewer 4 Comments

Point 1: Throughout the text, please distinguish between genes (abbreviations in ITALICS) and the proteins they code for. Genes don’t catalyze anything! This concerns DNA sequences versus amino acid sequences of “DTPSs” (cf. line 239: Pnl DTPS1, Pnl DTPS2, etc.).

Response 1: The suggested correction has been made throughout the text, where applicable.

Point 2: It is recommended to include the basic cyclization structures of oleoresin diterpenoids in the Additional Materials (stereochemistry!). This would certainly facilitate the reading by colleagues that are not absolutely familiar with this realm.

Response 2: An additional figure has been added to the supplementary materials, denoted as Figure S1 in the present revised version, illustrating the chemical structures of the most represented diterpenoids identified in Pinus spp. Such additional figure has been cited at lines 79-80 of the present revised version.

Point 3: Furthermore, it is strongly recommended to include the fragmentation patterns of compounds characterized by GC-MS and perhaps by comparison with literature data. Consider to include such “data that were not shown”, also to put them into Additional Materials?

Response 3: An additional figure has been added to the supplementary materials, denoted as Figure S4 in the present revised version, illustrating the mass spectra of the major diterpenoid compounds (nine DRAs) identified in Calabrian pine. Such additional figure has been cited at line 153 of the present revised version.

Point 4: What enzymes (or genes) are abbreviated with “LAS” (line 287)? This is usually used for lanosterol synthases?

Response 4: As reported in the manuscript (lines 275-278 of the present revised version), the abbreviation LAS is widely used to indicate the bifunctional DTPSs levopimaradiene/abietadiene synthase-type enzymes involved in DRAs biosynthesis in different conifer species. Moreover, Pc DTPS LAS1, Pc DTPS LAS2, Pb DTPS LAS1 and Pt DTPS LAS1 (line 287 of the original submission, corresponding to line 302 of the present revised version) are the original names used by the Authors to indicate the above type of enzymes identified in P. contorta (Pc), Pinus banksiana (Pb) and Pinus taeda (see Ro and Bohlmann, 2006; Hall et al. 2013).

Point 5: Is it correct to use only sequence similarities and deduce some biochemical activities based on literature data (cf. line 314 ff)? At least, the authors seem to recognize this problem when they discuss observations made with monoterpene synthases (line 334 ff). Indeed, a heterologous expression in yeast or bacteria would be necessary to clearly elucidate the specificities of encoded enzymes, although in planta helper proteins might be implied in cyclization reactions that would lack in such organisms.

Response 5: We are aware that the functional analysis of the full-length transcripts coding for DTPS by their heterologous expression in bacterial or yeast systems and in vitro enzyme assays would be crucial to elucidate the actual functions of P. laricio DTPSs, as also discussed at the end of the section 2.3 (lines 334-342 of the original submission). In this context, we are indeed planning to functionally characterize not only the complete transcripts encoding for the diterpene synthases but also those coding for the monoterpene synthases, that in the meantime we have already isolated and about to submit in a manuscript in preparation. It is important to remark here, however, that, unlike monoterpene synthases, in the case of diterpene synthases there seems to be no ambiguity in assigning the same function to proteins isolated from different Pinus species, providing they show a high level of homology among each other and belong to one of the four phylogenetic groups identified so far. On the basis of the work from Hall and Co-workers (2013), in fact, and apart from the proteins assigned to phylogenetic group 2, for which no biochemical activity was observed, the four proteins from three different Pinus species assigned to the phylogenetic group 1 seem to be all involved in the synthesis of abietane-type diterpene oleofins. Moreover, the two proteins from P. banksiana and P. contorta belonging to phylogenetic group 3 were found to produce isopimaradiene as the main product, while the two proteins from the same above species assigned to the phylogenetic group 4 were shown to produce pimaradiene. Taken together, and as a preliminary step, the above evidence led us to predict the potential functions of the DTPS proteins from Calabrian pine on the basis of the sequence relatedness of each of them with previously characterized pine DTPSs, taking into account phylogenesis. Such predictions of ours, moreover, were supported by the fact that the transcripts profiling of the Calabrian pine DTPS genes across different tissues was found to be consistent, as a whole, with that of the corresponding diterpenoids profiles, thus confirming the potential roles of the isolated DTPS genes presumed on the basis of their sequence homology.

Literature cited in the responses to Reviewer 4

Hall, D.E.; Zerbe, P.; Jancsik, S.; Quesada, A.L.; Dullat, H.; Madilao, L.L.; Yuen, M.; Bohlmann, J. Evolution of conifer diterpene synthases: diterpene resin acid biosynthesis in lodgepole pine and jack pine involves monofunctional and bifunctional diterpene synthases. Plant Physiol 2013, 161, 600–616. https://doi.org/10.1104/pp.112.208546

Ro, D.-K.; Bohlmann, J. Diterpene resin acid biosynthesis in loblolly pine (Pinus taeda): Functional characterization of abietadiene/levopimaradiene synthase (PtTPS-LAS) cDNA and subcellular targeting of PtTPS-LAS and abietadienol/abietadienal oxidase (PtAO, CYP720B1). Phytochemistry 2006, 67, 1572–1578. https://doi.org/10.1016/j.phytochem.2006.01.011

Reviewer 5 Report

The manuscript is interesting and valuable because it combines the analysis the particular compounds: diterpene resin acids with molecular identification and analysis of expression of their genes. Moreover, this approach provided new data on functional evolution of diterpene synthases in Gymnosperms.

The manuscript is well prepared and well written. The applied methods and the statistics are appropriate. The results are properly presented, and the conclusions are justified by the results.

I would suggest to add one figure to supplementary files (or even to the main text) – the chemical structures of the major identified compounds. Diterpene resin acids are quite specific compounds and they are not commonly known, so such an illustration would be valuable for the readers.

Line 497. Please write in full: Three-year old, without an abbreviation (Three-yr) which looks a little awkward in the beginning of the sentence.

Line 697. Please delete (R) since the other abbreviations of various plant organs and tissues are not used in this chapter.

Author Response

Response to Reviewer 5 Comments

Point 1: I would suggest to add one figure to supplementary files (or even to the main text) – the chemical structures of the major identified compounds. Diterpene resin acids are quite specific compounds and they are not commonly known, so such an illustration would be valuable for the readers.

Response 1: An additional figure has been added to the supplementary materials, denoted as Figure S1 in the present revised version, illustrating the chemical structures of the most representative diterpenoids identified in Pinus spp. Such additional figure has been cited at lines 79-80 of the present revised version.

Point 2: Point Line 497. Please write in full: Three-year old, without an abbreviation (Three-yr) which looks a little awkward in the beginning of the sentence.

Response 2: Done (line 519 of the present revised version).

Point 3: Line 697. Please delete (R) since the other abbreviations of various plant organs and tissues are not used in this chapter.

Response 3: Done (line 738 of the present revised version).
